# Breast Milk: A Source of Functional Compounds with Potential Application in Nutrition and Therapy

Cristina Sánchez [1], Luis Franco [2], Patricia Regal [3], Alexandre Lamas [3], Alberto Cepeda [3] and Cristina Fente [3,*]

1 Pharmacy Faculty, San Pablo-CEU University, 28003 Madrid, Spain; c.sanchez127@usp.ceu.es
2 Medicine Faculty, Santiago de Compostela University, 15782 Santiago de Compostela, Spain; luisfrancofente@gmail.com
3 Department of Analytical Chemistry, Nutrition and Bromatology, Santiago de Compostela University, 27002 Lugo, Spain; patricia.regal@usc.es (P.R.); alexandre.lamas@usc.es (A.L.); alberto.cepeda@usc.es (A.C.)
* Correspondence: cristina.fente@usc.es; Tel.: +34-600942349

**Abstract:** Breast milk is an unbeatable food that covers all the nutritional requirements of an infant in its different stages of growth up to six months after birth. In addition, breastfeeding benefits both maternal and child health. Increasing knowledge has been acquired regarding the composition of breast milk. Epidemiological studies and epigenetics allow us to understand the possible lifelong effects of breastfeeding. In this review we have compiled some of the components with clear functional activity that are present in human milk and the processes through which they promote infant development and maturation as well as modulate immunity. Milk fat globule membrane, proteins, oligosaccharides, growth factors, milk exosomes, or microorganisms are functional components to use in infant formulas, any other food products, nutritional supplements, nutraceuticals, or even for the development of new clinical therapies. The clinical evaluation of these compounds and their commercial exploitation are limited by the difficulty of isolating and producing them on an adequate scale. In this work we focus on the compounds produced using milk components from other species such as bovine, transgenic cattle capable of expressing components of human breast milk or microbial culture engineering.

**Keywords:** breast milk; infant formulas; functional compounds; milk fat globule membrane; breastmilk proteins; oligosaccharides; growth factors; milk exosomes; milk microbiome; probiotics

## 1. Introduction

Breast milk is an unbeatable food that alone meets the requirements of babies up to 6 months of age. The exclusively breastfed infants tend to have a satisfactory nutritional status. But the advantages of breastfeeding go beyond nutrition and are unanimously defended by all health establishments [1]. Among the innumerable benefits we can mention, in the neonatal period: lower mortality rates among breastfed infants exclusively during the first six months of life and improvement in the most prevalent pathologies in the first months of life (otitis media, asthma). As in the future life of the infant: babies who are breastfed have a reduction in dental malocclusion, lower risk of obesity, and even higher intelligence ratios [2].

According to the European Consensus on "Scientific Concepts of Functional Foods" [3] a food can be considered as functional if it is satisfactorily demonstrated to affect beneficially one or more target functions in the body, beyond adequate nutritional effects, in a way that is relevant to either improved stage of health and well-being and/or reduction of risk of disease. A functional food must remain food and it must demonstrate its effects in amounts that can normally be expected to be consumed in the diet. It is not a pill or a capsule, but part of the normal food pattern. The almost indisputable evidence scientist endorse breast milk as the best functional food, source of benefits for the infant and for the mother [4,5].

Knowledge of the composition of breast milk has highly increased and this will help understand the health benefits associated with breast feeding and to bring the composition of the infant formulas as close as possible to this "gold standard" [6]. Omics technologies, capable of detecting and identifying the set of molecules that exist in breast milk, have improved the understanding of their composition. This helps us to explain their physiological importance and their advantages from the point of view of infant health [7]. Moreover, the results of epidemiological studies and the growing knowledge of epigenetics allow us to understand the possible lifelong effects of breastfeeding [8]. Preventive medicine could also benefit from knowledge of the mechanisms by which human milk improves human development.

In this work we have compiled some of the components with clear functional activity that are present in human milk, and the processes through which they promote infant development and maturation as well as they modulate immunity. We summarize some of these compounds used as functional components for the development and/or improvement of infant formulas, any other food product, nutritional supplements, nutraceuticals or even for the development of new clinical therapies. The clinical evaluation of these compounds is limited by the difficulty of isolating and producing them on an adequate scale. Components from other species such as bovine, transgenic cattle capable of expressing components of human breast milk or microbial culture engineering, are used.

## 2. Methods

Web of Science—WOS—(CCC, DIIDW, KJD, MEDLINE, RSCI, SCIELO), PubMed, Cochrane databases, SCOPUS and Google Scholar were used as search engines for literature review, conducted for the period from January 1990 to January 2021. We included clinical trials, cohort studies, systematic (and non) reviews, and meta-analysis. The following keywords were used: breast milk, human milk, infant formulas, functional compounds, bioactive components, milk fat globule membrane, breastmilk proteins, human milk oligosaccharides, growth factors, milk exosomes, milk microbiome, probiotics, and their combinations. The search was not limited to title and abstract because our desired outcomes might have been mentioned in the full text of articles.

## 3. Functional components of the Breast Milk Fat Globule (MFG)

### 3.1. The Functional Structure of the MFG

Until recently, the concern of infant formula manufacturers had focused on mimicking as possible the energy and nutritional composition of human milk. However, the better characterization of human milk lipids and their interaction with other components, have driven current innovations in lipids in infant formulas [9]. Breast milk is a natural o/w emulsion in which lipid droplets, called fat globules, are biological entities secreted by mammary epithelial cells covered by a biological membrane, rich in bioactive substances, which is the interface with the intestinal tract. The secretion of MFG by the mammary epithelium comes within a diverse collection of proteins and lipids bound to the membrane in milk [10]. There is a broad scientific consensus that recognizes the importance of human milk fat globules in infant nutrition. The physical structure of the fat droplets may affect digestion, postprandial metabolism [11], and could even prevent from fat accumulation in adults [12]. Therefore, the knowledge of MFG microstructure serves as the basis for the adaptation of bioinspired functional emulsions in breast milk [10]. Baumgartner et al. developed an infant formula in which small lipid droplets are larger and have been emulsified with a polar lipid–protein interface that simulates MGF membrane (MFGM). These emulsions improve fat digestion compared to lipid droplets wrapped solely in proteins [11]. Preclinical studies suggest possible long-term benefits on body composition. Although the mechanisms remain unclear, animal studies showed that early nutrition is associated with sustained effects on obesity in later life (Ronda et al., 2020). A clinical trial is currently underway to test Nuturis® (NCT01609634; trial of new infant formula in healthy subjects on growth, body composition, tolerance and safety).

The interactions and possible synergies between the different components of MFGM are still not well understood, but the best results in preclinical and clinical trials, particularly with regard to infections and neurodevelopment, indicate the joint addition as MFGM is more interesting. The addition of MFGM to infant formula is a safe and justified improvement strategy for infant formulas, bringing them closer to the nutritional profile of human milk [13]. The membrane fraction is an inherent component of all mammalian milk, however its biological value is lost in infant formula due to the use of vegetable oils to adjust the fat composition. The addition of this enriched milk fraction may be closing a gap that was lost in the switch to vegetable oils for infant formulas [14]. Different commercial preparations of bovine MFGM, from serum or cream concentrates, with considerable variations in their composition, are available for infant formulas [15,16]. Clinical trials have demonstrated the benefits of its introduction in these commercial preparations. Table 1 shows some of the different commercial alternatives and the improvements that their addition seeks.

### 3.2. MFGM Lipids

Glycerolipids (phosphatidylcholine, phosphatidylethanolamine, phosphatidylinositol, phosphatidylserine) and sphingolipids (sphingomyelin and gangliosides) are complex lipids with amphipathic nature and are present in not so important quantity in MFGM, but they are structural components with very interesting functional properties [14].

The phospholipids of MFGM are a source of choline, an essential nutrient involved in various biological processes, mainly metabolism, but also in the construction of membranes in the brain and nervous tissue. Newborns require large amounts of this compound for the quick growth of organs and the biosynthesis of cell membranes [17]. EFSA considers that 130 mg per day is the adequate intake of choline for the first six months of life [18].

About half of the complex lipids in the fat globule membrane are sphingolipids. The digestion of the main sphingolipid in breast milk, sphingomyelin, generates ceramide, sphingosine, and sphingosine-1-phosphate, with numerous signaling functions mediated by intracellular pathways, whose effects are related to the regulation of cell growth, differentiation, apoptosis, and the migration of immune cells [19].

Gangliosides consist of a hydrophobic ceramide and a hydrophilic oligosaccharide chain that carries one or more sialic acid residues, in addition to various sugars such as glucose, galactose, N-acetylglucosamine, and N-acetylgalactosamine. Although gangliosides were initially isolated from the brain and are especially abundant in neural tissues, they are widely distributed in most vertebrate tissues and fluids. Breast milk, very rich in these compounds, significantly increases total gangliosides in the intestinal mucosa, plasma and brain. Therefore, they possibly play an important role in the development of the infant's tissues, especially the small intestine, improving their oral tolerance [20]. Clear differences in the concentration and type of gangliosides have been observed in mothers of premature babies and also during lactation: disialoganglioside is abundant in colostrum while monosialoganglioside accounts for 85% of total gangliosides in mature milk. These changes suggest different functionalities according to the moment of the child's development [21]. The importance of gangliosides in immunity and protection against infection has been extensively studied. Some studies suggest a contribution in the processes of proliferation, activation and differentiation of immune cells, and the immunomodulatory effect could be more prevalent in early lactation when the milk contains the highest amount of disialoganglioside [22,23]. These compounds are true probiotics, the growth-promoting effect of bifidobacteria and lactobacilli is well-known, [24]; as well as the protective effect against Giardia muris and Giardia lamblia [25]. The benefits of these sialic compounds in the development and maturation of the newborn have recently been summarized [26]. Besides, the modification of the physical properties of the brush border membrane induced by gangliosides in breast milk could lead to an improvement in the absorption of polyunsaturated fatty acids ω3 and ω6 in contrast to saturated fatty acids [27].

### 3.3. MFGM Proteins

More than 100 different MFGM proteins have been identified. All of them are secreted by a process that is unique to mammary epithelial cells [28]. Some will be weakly adhered to the membrane, like in the case of lactoadherin; others, like mucin, have a peripheral distribution; while butyrophylin is more integrated in the fat globule. The more peripheral proteins can account for a significant percentage of the membrane weight when the fat globule size is small, in the early infancy, or longer in time when birth is premature [29]. Butyrophylin, on the other hand, is more abundant when the size of the globule increases. Moreover, most of the proteins of the fat globule are highly glycosylated and, probably, this is the reason why they are not altered by the low pH and the activity of pepsin in the baby's stomach, thus maintaining their functionality in the baby's intestine. Thereby, for example, mucin and lactoadherin are important for intestinal development, as they help stimulate the health of the intestinal epithelium showing antiviral and antibacterial activities in the infantile gastrointestinal tract; butyrophyllins have been related to the regulation of the immune response; and activated lipase by bile salts is related to the digestion of lipids [4].

### 3.4. Fat Globule Nucleus' Lipids

The nucleus of the MFG is made up of triglycerides (TAG) containing more than 200 different fatty acids. Most are found in very low and variable concentrations in mothers and throughout lactation, but always presenting a high content of palmitic, oleic and linoleic acids [30]. These three fatty acids occupy highly conserved positions in the TAG. The first, strongly concentrated in the sn-2 position, oleic in the sn-1 position and in the sn-3 position, the linoleic. The different fatty acids present in human milk can come from the synthesis of new fatty acids in the liver or breast tissue, from the mobilization of endogenous fats, or from the diet [31]. With a low-fat, high-carbohydrate diet, there are higher amounts of medium chain saturated fatty acids in breast milk, as a result of their increased synthesis in breast tissue [32]. The variety of fatty acids consumed in the diet during pregnancy and lactation are key factors, as they determine the transfer of fatty acids through the placenta and then through breast milk. It has been shown that it changes during the lactation period and reflects the diversity of food situations around the world and even the different body conditions of mothers. We can conclude that the fatty acid pattern directly relates the maternal eating pattern with that of the child [33]. The specific $\beta$ position of palmitic acid, the most common saturated fatty acid in breast milk, improves their absorption and prevents the formation of soaps. This improves absorption of macroelements such as calcium and magnesium, reduces constipation, and improves the intestinal well-being of the breastfed infant [34]. But the benefits of $\beta$-palmitate also include homeostasis of intestinal mucosa, intestinal microbiome, and the newborn's immune response [35]. However, the TAG structure is different in the vegetable oils used in infant formulas. Given its benefits, it seems logical to include TAGs in which palmitic is occupying the position of most difficult hydrolysis ($\beta$ position). This improvement have clinical evidence [36] and is included in most infant formulas (see Table 1).

The essential $\alpha-$linolenic and linoleic acids, very important for the growth and maturation of the baby's brain, are also present in breast milk. The long-chain polyunsaturated fatty acids (PUFA), synthesized from them, represent approximately 15% of the total lipids in breast milk and must be of great importance in child development, since during the perinatal period they accumulate in considerable quantities in certain tissues such as the central nervous system, particularly in the synaptic neuronal membranes, and in the photoreceptor retina cells. They have been studied for their benefits in development, their cardioprotective role, and their biological anticancer, anti-inflammatory, and antioxidant functions [37]. However, the enzyme systems in the young child are still immature and the degree of synthesis may not be sufficient to meet the high needs during this stage, so these fatty acids become, in practice, essential. Fatty acid profiles in blood with an excess of linoleic acid and a deficiency of whole blood docosahexaenoic acid (DHA) and arachidonic acid (ARA) have been associated with an increased risk of developing bronchopulmonary dysplasia,

retinopathy of prematurity, and sepsis [38]. Supplementation of infant formulas with ARA and DHA is a common practice. Intervention studies have shown that PUFA supplementation has positive developmental outcomes and causes formula-fed infants to show results regarding cognitive function, visual clarity, and immune response similar to those given breast milk (Table 1).

**Table 1.** Examining lipid breast milk components used in health applications, summary of evidences (clinical trials, cohort studies, and meta-analysis).

| Component | Utilization | Health Effects | References |
|---|---|---|---|
| MFG structure (Nuturis®) | Infant formula | Improves lipids' postprandial digestion and metabolism. Prevents from fat accumulation in adults. | [11,39] |
| Bovine serum derived MFGM (Lacprodan-10 and other not specified) | Infant formula | The MFGM supplemented formula can decrease the artificial nutrition's metabolic firm. General phenotypes observed to be more similar to BF group. Similar results to natural lactation as to fecal metabolome and microbiome. Better cognitive results than the control group at 12 months. Lesser incidence of Acute Otitis Media and lesser use of antipyretic drugs. Similar results of febrile and diarrhea episodes to natural lactation. Lower levels of HDL cholesterol and homocysteine. | [13,28,40–47] |
| Bovine serum derived MFGM | Complementary nutrition | Lower diarrhea prevalence. Lower levels of IL-2 proinflammatory cytokine. Higher serum levels of choline. Higher levels of circulating amino acids and better anthropometry. | [48,49] |
| Bovine serum derived MFGM (Lacprodan-10) and lactoferrin | Infant formula | Significantly improved performance in Bayley's test at 12 months of age. Long-lasting effects of nutritional intervention at 18 months. Lesser incidence of respiratory and gastrointestinal adverse effects. | [50] |
| Non specified MFGM and other bioactive components | Infant formula | Visual function (measured through latency and amplitude) was significantly improved. | [51] |
| Bovine milkfat derived MFGM (Inpulse® and other not specified) | Infant formula | Lowers incidence of febrile episodes and reduces the number of days with fever of children between 2.5 and 6 years. | [52] |
| Complex milk lipids | Whole added milk | Lower duration of diarrhea caused by rotavirus in infants between 8 and 24 months. | [53] |
| β-palmitate | Infant formula | Increases bioavailability of palmitate and calcium. Softening of stool. | [36] |
| ARA and DHA | Infant formula | Lower incidence of upper and lower respiratory tract infections. Lower incidence of diarrhea. Beneficial effects on immune system of developing infants. Lower incidence of allergies and asthma. Improves visual and cognitive functions. | [54–62] |
| MCFAs | Infant formula for premature babies | Lesser intestinal colonization by Candida | [63] |

Finally, the short and medium chain fatty acids (MCFA) found in breast milk may have nutritional advantages due to their absorption (directly into the portal circulation) and faster metabolism. Absorption is almost complete at different concentrations, so they would have benefits in conditions of limited fat absorption, for example, in premature babies [64]. At present, formulas are being prepared with structured lipids which contain essential fatty acids and medium chain fatty acids with easy absorption characteristics and that mimic the structure of the TAGs of the breast milk [65].

## 4. Human Milk Oligosaccharides (HMOs)

In recent years, non-nutritive carbohydrates in human milk have attracted considerable attention: they are the third component in percentage after lactose and lipids, practically equal to proteins, and are being investigated for their use as functional components. A baby that consumes 800 mL of milk would take 10 g of oligosaccharides per day [66]. The synthesis of HMOs is very energy-expensive for the mother and this can only be understood taking into account the human reproductive strategy of a large contribution from the parents to raise relatively few babies for a long period. Breast milk is unique in its diverse and complex composition of HMOs.

Up to 200 different HMOs have been identified, which can contain five different monosaccharides (glucose, galactose, N-acetylactosamine, fucose, and sialic acid). They all carry lactose at their reducing end, which can be fucosylated or sialylated into small HMOs, or elongated with disaccharides to form larger HMOs ranging in size from 3 to 32 sugars [5]. They are different from those found in the milk of any other mammal; bovine milk, the base for most infant formulas, is 1000 times lower in concentration [67].

The structures of HMOs vary according to maternal genetics, consequently, the composition of HMOs in the milk of women varies significantly [68]. In addition to genetics, other maternal factors such as age or diet cause different composition profiles in HMOs. These differences have been related to the microbiome of breast milk, which could even be predicted by knowing its HMOs [69].

Since the oligosaccharide composition orchestrates the development of the infant's intestinal microbiota, it could be related to short-term infant health outcomes, but it could also have long-term consequences for health status and risk of disease later in life (Bode 2015). HMOs can be considered as constituents of an innate immune system by which the mother protects her baby through breastfeeding [70].

Once ingested, the HMOs from breast milk reach the distal area of the small intestine and colon practically intact. They are recognized probiotic agents (the first in the human diet) that stimulate the growth of beneficial microorganisms, mainly of the genus Bifidobacterium (dominant species in breastfed infants) and, to a lesser extent, some strains of Bacteroides and Lactobacillus. As these bacteria specifically express sialidases and fucosidases, it is believed that HMOs select these strains, since other bacteria are not capable of using them [71]. On the other hand, maternal HMOs increase the adhesion of the selected strains to the intestinal mucosa, improving their persistence in the mucosa and increasing the anti-inflammatory effects on the human intestine [72].

But in addition, HMOs can protect infants by reducing the incidence of bacterial, viral, or parasitic intestinal diseases, acting as antiadhesives in interactions with the host, in two ways: selectively binding to pathogens or their toxins, then inhibiting its adherence to glucan ligands on the mucosal cell surface [73] or they can bind themselves to glucocalyx on the surface of epithelial cells [5]. These effects are complemented by their ability to compete with viruses for C-lectin antigen uptake receptors on dendritic cells [74]. *Entamoeba histolytica* (a very common protozoan parasite in many areas of the world) also expresses a lectin, which is an important virulence factor involved in binding to intestinal epithelial cells. Only HMOs with a terminal galactose are able to compete and therefore be active [5].

The partial metabolization of HMOs gives rise to "postbiotic" compounds that stimulate the growth of other types of butyrate- and propionate-producing flora. These short-

chain fatty acids have a trophic effect on the intestinal barrier, stimulating mucin release and modulating the immune system, promoting immune tolerance [75]. These strains also keep the growth of potentially pathogenic bacteria under control by reducing the nutrients available for potentially harmful bacteria. A direct action of HMOs in breast milk on common bacteria in pregnant women and young children has been postulated. This bacteriostatic action has been demonstrated in the case of group B Streptococcus, which cannot proliferate in a medium with sialylated HMOs. This action is synergistic with antibiotics and extends the therapeutic utility of these molecules [76].

But HMOs not only affect directly the microorganisms, they also act by altering the host cell responses, causing changes in the glucocalyx of epithelial cells [77] or modulating the maturation of the intestinal epithelium. In addition, they affect immune cells, reducing the expression of pro-inflammatory cytokines [78] or showing different effects of activation and inhibition of Toll-like receptor (TLR) signaling pathways. Thus, some maternal HMOs can inhibit TLR receptors involved in pathologies such as cystic fibrosis or systemic lupus erythematosus. This could be the basis for the use of some of these compounds for the therapeutic approach of these pathologies with nutritional contribution [79].

Although most of the benefits of HMOs are due to their local effects in the intestines of babies, in recent years it has been shown that some are absorbed, as such or partially metabolized, and pass into the systemic circulation. In this way they could exert their effects beyond the intestinal lumen and the mucosal surfaces of the intestine. Thus, the blocking of the binding of microbial pathogens to cell surface receptors has been described not only in the intestine, but also in other sites such as the urinary tract [80] or the respiratory tract [81].

The benefits of HMOs extend beyond the lactation period. Some studies suggest that they could improve cognitive function. Sialic acid residues that come from these HMOs improve brain development in animal models [82,83]. Other studies show a significant association with decreased allergy risks [84]. Regarding benefits of the most predominant HMOs, correlations have been established between their concentration and the incidence of diseases such as diarrhea, necrotizing enteritis, or respiratory or urinary infections [85].

Knowing the benefits that HMOs provide to the infant, they have begun to be introduced as functional components in infant formulas. However, human milk contains up to 200 very different oligosaccharides, in monosaccharide components and in size [67]. Introducing one or even more HMOs into formulas is unlikely to be sufficient to fully mimic all the beneficial effects associated with the complexity of the composition of human milk. Furthermore, we must not forget that human milk is not a static fluid and the expression of HMO changes with the stage of lactation, maternal genetic or anthropometric factors, and even with geographical location, among other factors [86,87].

Since the beginning of this century, complex oligosaccharides such as galactooligosaccharides (GOS), inulin-type fructans or their combination, or with mixtures of polydextrose (PDX), and acid oligosaccharides (AOS) have been introduced into infant formulas [88]. The benefits, but also some negative aspects, of this supplementation have been highlighted [89,90]. Therefore, the EFSA Panel considers that there is insufficient evidence of beneficial or adverse effects on infant health of these types of fiber added to infant and follow-on formulas [6,91].

The oligosaccharides in the serum permeate of cow and goat milk, although less diverse and in less quantity than HMOs [92], represent attractive alternatives [90]. To approximate its composition to HMOs, especially in terms of fucosylated oligosaccharides, chemoenzymatic processes have been used on these by-products of the dairy industry [93]. Chemoenzymatic techniques, together with microbial metabolic engineering processes, are currently used to produce HMO in sufficient quantities to be able to address the design of functional and nutraceutical foods, in addition to preclinical and clinical studies to evaluate them [94,95]. However, to emulate the biological function of HMOs, structure–function relationships must be taken into account, which is a major challenge for the biotechnology industry [96]. Although the synthesized HMOs show a structural identity with the natural

ones, the introduction of these ingredients requires the evaluation and approval of the different administrations in the world. In Europe they must be designated as novel foods by EFSA and in the US they must receive the GRAS rating from the Food and Drug Administration (FDA). There are several of these synthetic HMOs already approved on the market and some of them (2′-fucosillactose (2′-FL) and lacto-N-neotetraose (LNnT)) incorporated in commercialized infant milk, with good results in terms of safety, tolerance, and intestinal benefits in intervention studies [97–99]. They are also beginning to be incorporated into products for a wider audience [96]. More HMOs are expected to be added to the international market as the different administrations authorize them. This will improve pediatric nutrition but may also be an opportunity to expand its use in nutraceutical applications and improve health in human populations. Table 2 shows some of the products that are already being used.

**Table 2.** Examining human milk oligosaccharides (HMOs) breast milk components used in health applications, summary of evidences (clinical trials, cohort studies, and meta-analysis).

| Component | Utilization | Health Effects | References |
|---|---|---|---|
| 2′FL | Infant formula | Comparable growth to babies that were given natural lactation. Lower incidence of respiratory and intestinal diseases. | [100,101] |
| 2′FL/LNnT | Nutritional supplement for adults | Improves intestinal flora (Bifidobacterium spp.) in patients with Irritable Bowel Syndrome. | [102] |
| 2′FL/LNnT | Infant formula | Softened stool. Lesser nocturnal awakenings. Lesser bronchitis, lower respiratory tract infections, use of antipyretics and antibiotics. Similar metabolic firm to babies that were given natural lactation. | [97,98] |

## 5. Breast Milk Proteins

Human milk proteome is very complex, with more than 1500 different compounds, 524 of which had not been previously described and their functions are unknown [103]. In addition to the lower proteolytic capacity and the higher permeability of the infant's intestinal tract epithelium, as we see, the proteins in breast milk not only support the infant's growth, but also fulfill many other functions that ensure the maturation of the organs and systems and provide protection against specific deficiencies and pathogens. Of the total proteins, 83 are differentially abundant. Furthermore, some of the peptides released in their partial digestion also show bioactivity [104,105]. In the following section, we summarize some of these compounds used as functional components.

### 5.1. Caseins

As for caseins, their concentration is the lowest of all species, corresponding to the slow growth rate of human infants [106]. We can find small amounts of αS1-casein and, to a greater extent, β- and κ-casein, but not αS2-casein, present in bovine milk [4]. K-casein stabilizes insoluble α- and β-caseins, forming a colloidal suspension (the casein micelle).

K-casein, as well as its glycomacropeptide hydrolyzate (GMP), are bound to complex oligosaccharides that, similar to mucosal glucans, can bind to pathogens [105]. GMP contains very low levels of phenylalanine and is used in food products for the nutritional management of children with phenylketonuria [107,108]. Beyond its nutritional utility, GMP has been tested to reduce glycemic response with good results in adults [109]. On the other hand, GMP can act as a bifidogenic factor [110].

The digestion of caseins produces peptides with a wide variety of relevant properties. The inhibitory effect of the angiotensin-converting enzyme by various casein-derived peptides is known, including the peptide known as C12 (FFVAPFEVFGK), which demonstrated a significant reduction in systolic and diastolic blood pressure [111]. It is currently available commercially by the company DMV International (Veghel, The Netherlands), for the elaboration of nutraceuticals.

Angiotensin II, generated by angiotensin converting enzyme (ACE), can promote oxidative stress and neuroinflammation, leading to neurodegeneration and brain aging [112]. Centrally active ACE inhibitors and angiotensin receptor blockers have been suggested to reduce the risk of Alzheimer's or delay its progression, regardless of their blood pressure lowering effect [113–115]. The antihypertensive tripeptide Met-Lys-Pro (MKP), derived from bovine casein, has been tested in animal models showing potential as a therapeutic agent for cognitive function [116]. Recently, the study by Yuda et al. [117] showed that MKP supplementation may have the potential to improve cognitive function in adults without dementia, with good tolerability and no treatment-related side effects, during the 24 weeks of treatment and 2 weeks after treatment. Other peptides derived from casein and with a high proportion of proline, such as Colostrinin®, present in human, bovine, and caprine colostrum, modify the expression of the genes involved in the synthesis of the b-amyloid protein, increase the expression of proteases that eliminate the accumulation of the b-amyloid protein and improve inflammatory and oxidative damage. Colostrinin® has shown a psychostimulatory effect in animals and humans. This peptide has been successfully tested in Alzheimer's patients, showing demonstrable clinical effects without side effects [118–122]. A Colostrinin® nutraceutical (ReGen Therapeutics Ltd., London, UK) is currently available for use in neurology and degenerative diseases. See Table 3.

### 5.2. Serum Proteins

Serum proteins include, but are not limited to: α-lactalbumin, lactoferrin, secretory immunoglobulin A, lysozyme, and osteopontin.

#### 5.2.1. α-lactoalbumin

The α-lactalbumin is a serum protein that makes up more than a third of the total protein in human milk, unlike cow's milk in which it is found in much lower amounts [123]. It has prebiotic activity on Bifidobacterium [124]. Furthermore, its partially digested peptides also have biological activity. They bind to calcium, iron, and zinc, which increases their absorption and exerts an antimicrobial action mainly against Gram-positive bacteria [125]. These biological activities have also been verified with bovine α-lactoalbumin added to infant formulas [126].

But most uses for α-lactalbumin stem from its amino acid composition. Tryptophan, is a precursor to the neurotransmitter serotonin that has been linked to central nervous system functions such as appetite, sleep, memory and learning, regulation of temperature, mood, behavior, and maturation of neurons and synaptic connections. Cysteine, a sulfur amino acid that stimulates the production of glutathione that plays an important role in protecting cells against oxidative stress; and the branched-chain amino acids. Leucine, isoleucine and valine, stimulate postprandial anabolism of muscle proteins [123]. For all this, its addition to infant formulas makes them more similar to breast milk and supports infant development and growth. But it is also used in adult supplements to protect muscle mass in older adults or athletes and to modify neurological or behavioral outcomes such as sleep and mood [127–130].

In addition, it may have clinical applications in cancer therapy or to improve immune function. Beyond its functional activity in pediatric nutrition, the α-lactoalbumin in human milk, known as HAMLET (α-lactoalbumin made lethal to tumor cells), exhibits anticancer activity in around 50 different types of cancer cells for which it has been tested [131]. Microbial cultures [132] and transgenic animals [133] are used for large-scale production of the human variant of α-lactoalbumin. HAMLET has been used successfully in oncology:

instilled in situ before resection of tumors in the bladder, improving the prognosis of this type of cancer [134]; in topical application in papillomas, achieving complete remission of lesions in more than 90% of treated patients and without significant side effects [135].

Another possible therapeutic application of $\alpha$-lactoalbumin is in the treatment of epilepsy. This protein has been shown to be effective in some animal models of epilepsy and epileptogenesis [136–138] and in patients with epilepsy [139,140].

### 5.2.2. Lactoferrin

Lactoferrin is a multifunctional glycoprotein that occurs in high concentrations in colostrum and, although in lower concentrations, in mature milk. It plays a very important role in protection against pathogens (bacteria, fungi, and viruses) and in immune regulation [139,140].

The antimicrobial effect was the first protective activity of lactoferrin to be identified. It is a protein of the transferrin family that has a low degree of iron saturation, which makes it retain the necessary iron as a growth factor for pathogens, hence its bacteriostatic effect. Iron sequestration has also been related to the prevention of biofilm formation [141] and its activity against fungi [142]. Its bactericidal action does not depend on iron but on its direct interaction with the lipoteichoic acid of Gram-positive bacteria or the liposaccharides of Gram-negative bacteria. The action of lactoferrin against Gram-negative bacteria is synergistic with lysozyme, which is also present in breast milk in relatively high concentrations. Complexes formed with the bacterial lipopolysaccharide (negatively charged) create holes in the outer membrane through which lysozyme penetrates the outer membrane, thus gaining access to and degrading the proteoglycan matrix, resulting in bactericidal action [143]. The amebicidal action of lactoferrin is also explained by its binding to the lipid membrane of the parasite, which causes its alteration and damage [144]. The antiviral activity of lactoferrin has also been studied a lot, but above all these are in vitro studies and not so many clinical trials. Some of the mechanisms have been identified, such as the inhibition of viral adhesion to the host cell, preventing the virus from entering the cell, through direct binding to the virus surface or the removal of iron [145]. But lactoferrin has also been found in the cell nucleus, so its action could also be intracellular [146]. Early breastfeeding has recently been postulated, due to its high lactoferrin content, as protection against coronavirus infection [147].

The direct antimicrobial properties of lactoferrin are complemented by immunomodulatory properties due to its ability to interact with numerous cellular and molecular targets. At the cellular level, it modulates the migration, maturation, and functions of immune cells. At the molecular level, in addition to iron binding, interactions with the cell surface explain its modulating properties [148]. Lactoferrin has shown to be a promising agent for reducing prevalent pathologies in neonates [149].

Being an iron-binding glycoprotein, it is hypothesized that lactoferrin plays a key role in the homeostasis of this mineral. A specific receptor for lactoferrin has been detected in brush border cells, which would explain the greater efficiency in the absorption of this mineral [150].

Lactoferrin is very similar between species, and the homology between human and cow's milk is 77%. The purification of bovine lactoferrin [151] has allowed the use of bovine lactoferrin in infant formulas and this supplementation provides multiple benefits in the prevention of infections in newborns, premature and term, as well as in the reduction of morbidity and mortality [141]. As examples of benefits shown in clinical trials in infants, we can cite: the content of total body iron and the intestinal absorption of iron in the intestine in babies increases significantly when they are fed with formula milk fortified with bovine lactoferrin [152]; it was effective in reducing the risk of infection during the first 12 months of life [153]; it reduced the incidence of bacterial sepsis in very underweight preterm infants in the first 45 days of life by 70% compared to placebo treatment in a large randomized study [154]; also the incidence of fungal sepsis in premature infants was reduced with supplementation with bovine lactoferrin [155]; prophylactic administration

reduced both the duration and severity of diarrhea for 6 months [156]; more recently, the consumption of bovine lactoferrin in the first 10 days of life has been directly associated with less late-onset sepsis, less necrotizing enterocolitis, and lower mortality [157].

Large-scale production of recombinant human lactoferrin, in *Aspergillus awamori* (Agennix, Houston, TX, USA), rice (Ventria Bioscience, Sacramento, CA, USA), and transgenic cows (Pharming, Leiden, The Netherlands), has allowed the performance of multiple clinical trials that demonstrate its potential as an antimicrobial agent with clinical efficacy in the treatment of infectious diseases in humans. Recombinant human lactoferrin has been shown to act in vitro against pathogens such as *E. coli*, *Staphylococcus aureus*, *Pseudomonas aeruginosa*, *Helicobacter pylori*, *Bacillus subtilis*, *Vibrio cholerae,* and *Candida albicans* [158]. But it also inhibits the in vitro replication of human cytomegalovirus, HIV, herpesvirus, hepatitis B and C, hantavirus, human papillomavirus, rotavirus, adenovirus, and influenza A [146]. Preterm infants who receive it, show a trend toward lower infectious morbidity and changes in the fecal microbiome [159,160]. However, more trials are needed to demonstrate the efficacy of lactoferrin supplementation for the treatment of sepsis and necrotizing enteritis in preterm infants [161,162]. The efficacy of recombinant human lactoferrin supplementation was also demonstrated in adults: reducing mortality in adults in intensive care due to severe sepsis [163]; increasing the efficacy of *H. pylori* eradication therapies [164], decreasing postantibiotic diarrhea in elderly patients [165]; increasing the efficacy of standard interferon (IFN) and ribavirin therapy in hepatitis C and other viral infections [166] or reducing the symptoms and duration of the common cold [167].

Anti-inflammatory, antioxidant, immunomodulatory, and antitumor activities are known in vitro [168] and in animal studies [169]. Additionally, it can be internalized into the cell nucleus, the site of action of most anticancer drugs, and has been used as a targeting ligand to achieve active delivery of anticancer drugs to tumor tissue [170]. Regarding clinical trials: oral consumption of 3 g/day bovine lactoferrin significantly affected the growth of adenomatous polyps in the colon [171]; administration of human recombinant lactoferrin increased survival, in a randomized double-blind placebo-controlled study, in an average of 65% of patients with advanced stage non-small cell lung carcinoma [172]; it also showed marked improvements in overall survival as an adjunct to standard chemotherapy in patients with newly diagnosed lung cancer [173] and with breast cancer [174].

5.2.3. Human Milk Secretory Immunoglobulin A (IgAs)

For IgAs and lysozyme, protection of the infant against pathogens is the only significant bioactivity [175]. Human colostrum can contain antibody concentrations of up to 12 g/L. In mature milk, this ability to provide robust protection against pathogens is preserved with approximately 1 g/L of milk immunoglobulin [176]. IgAs present in human milk is resistant to protein digestion and is found intact in the intestines of breastfed babies. Its action against pathogens is based on immune exclusion that mainly involves T-cell-dependent monoclonal antibodies with high specificity toward the pathogen's surface antigens. But also, it has a non-specific antipathogenic activity promoting the initial development of the newborn's microbiota. Although the intestinal mucosa is capable of producing IgAs, the amount found in the intestine of naturally fed children exceeds that of formula-fed children. In addition, the infant receives antibodies against the antigens to which her mother has been exposed [105,177].

Repeated inoculation of an antigen into gestating dairy cows can stimulate increased production of high levels of colostral immunoglobulin against that target antigen, resulting in hyperimmune bovine colostrum (HBC). HBC has been used successfully to treat diarrhea in children [178] and for specific antimicrobial prophylaxis in a cohort of healthy adults [179]. It has also demonstrated its efficacy in the treatment, or as a preventive element of potentially fatal pathogens, in vitro and with animal models, such as *C. difficile*, *Cryptosporidium*, *E. coli*, *Shigella*, rotavirus [180–182], and AIDS virus [183]. HBC is avail-

able in Australia as a tablet (Travelan®) for the prevention of traveler's diarrhea (Anadis, Campbellfield, Victoria, Australia).

### 5.2.4. Osteopontin

It is a highly glycosylated and phosphorylated protein whose concentration in breast milk is relatively high (140 mg/L) [184] and its role has been studied not only in bone remodeling, but also in the modulation of inflammation and immune function [104].

Osteopontin may act as a carrier for lactoferrin and thus cause other immunomodulatory proteins to further enhance immune competence [185].

Although it is not currently added routinely to formulas, the addition of osteopontin to infant formula, at concentrations similar to human milk, resembles the innate and adaptive immune responses to those of naturally fed babies. Thus, at four months of age, babies breastfed naturally or fed a formula supplemented with osteopontin, have fewer episodes of fever and a less pro-inflammatory immune response (they showed lower levels of pro-inflammatory TNF-$\alpha$ and higher levels of interleukin-2) compared to the standard formula group [105]. Higher TNF-$\alpha$ levels in formula-fed infants, compared to breastfed infants, have been interpreted as a pro-inflammatory immune response to early formula feeding [186]. The immunomodulatory function of osteopontin has been confirmed in other clinical trials [187].

Other neurodevelopmental functions of this protein have been highlighted in animal trials [188].

### 5.2.5. Bile Salt Stimulated Lipase (BSSL)

In infants, the digestion of triglycerides is carried out by gastric lipase, pancreatic lipase, and the BSSL, a very abundant lipase in human milk. The BSSL compensates for the limited capacity of pancreatic enzymes in the first months of life. This lipase is inactive until the chyme reaches the duodenum and comes into contact with bile salts, hence its name. Its lipolytic activity has been demonstrated against cholesterol esters, fat-soluble vitamin esters, galactolipids, and ceramides [189]. Pasteurization of human milk inactivates it, which explains the lower weight gain of infants fed with heat-treated donor milk [190]. This enzyme is not detected in cow's milk. The option for incorporation into infant formulas is recombinant bile salt-stimulated lipase, produced by human cell culture. The addition of recombinant human BSSL to pasteurized breast milk or infant formula improves growth rate and the absorption of long-chain polyunsaturated fatty acids in premature infants [191] and small for gestational age infants [192]. Apart from its effects on nutrition, BSSL has been shown in vitro to act as a decoy receptor for human calicivirus strains and may provide some protection against gastroenteritis due to norovirus infection in infants [193]. Furthermore, it can bind to specific dendritic cells and reduce the risk of HIV infection in vitro [194].

## 6. Non-Protein Nitrogenous Compounds

The non-protein nitrogen fraction represents approximately 25% of the total nitrogen and comprises many bioactive molecules. As we have said before, 10–15% are endogenous peptides and the remainder comes from urea, nucleotides, carnitine, creatine, free amino acids, DNA, and RNA [195].

### 6.1. Free Amino Acids (FAAs)

In addition to proteins, human milk contains free amino acids (FAAs), which are a good source of nitrogen for the infant. They are directly available for absorption without prior digestion. The protein-bound form, for each individual AA, decreases during lactation, in correlation with the baby's protein needs for growth [196]. On the contrary, the levels of FAAs show dynamics during lactation that are highly specific for each AA: while the levels of some FAAs decrease in the first 3 months of lactation, others remain stable or increase sharply. Surprisingly, these dynamics of FAAs during lactation are a con-

stant worldwide. This fact suggests that they are tightly regulated throughout infancy and, consequently, that they may have specific functions in the developing neonate. Thus, glutamine, glutamate, glycine, serine, and alanine, constantly increase in the first 3 months of lactation. The levels of most other FAAs remain relatively stable throughout infancy [197].

Glutamate and glutamine are the most abundant FAAs. Gestational age is a determining factor in the levels of free glutamine in breast milk. In the first month of lactation, the levels of free glutamine in the milk of premature mothers are almost three times lower than those observed in full-term milk. The levels of all other FAAs do not vary with gestational age [198]. In newborns, dietary glutamine and glutamate are trophic factors of intestinal epithelial cells. Therefore, they will improve the function of the intestinal barrier and influence the development of immune cells. They have also been found to exert anti-inflammatory effects and modify the intestinal microbiota, which could play a role in allergic sensitization [199]. The findings that relate glutamate and glutamine levels with child anthropometry are also surprising, they are positively associated with increased height and weight in infants [200]. Along these lines, it has been found that milk for boys (who gain more weight and height in this period) tends to have higher levels of free glutamine and glutamate than milk for girls in the first 3 to 4 months of lactation [201]. The estimated free glutamine intake in breastfed infants is up to 4.5 times higher than the acceptable daily intake (ADI) established by the EFSA [6] for infants, therefore some authors believe that, since there is no reason to assume that breast milk feeding is unsafe for infants, setting an ADI below the normal intake range with a safe diet such as breastfeeding seems inappropriate [202].

Breast milk is not only adapted to the nutritional and immunological needs of the infant, its composition also varies throughout the day. Circadian fluctuations in some bioactive components, such as tryptophan, transfer chronobiological information from the mother to the child to aid the development of the biological clock [203]. Could we think of a future in infant formulas formulated by sex and for day and night?

Formula-fed infants exhibit faster weight gain, a different fecal microbial profile, as well as elevated levels of serum insulin, insulin growth factor 1 (IGF-1), and branched-chain amino acids. Since infant formula contains more protein and fewer free amino acids than breast milk, these are believed to be key factors that explain phenotypic differences between infants. A preclinical study in Rhessus monkeys fed low-protein and high-FAA formula, similar to breast milk, has advanced that, although the growth and metabolic performance of infants was more similar to naturally fed infants, it was not enough to reverse the accelerated growth and specific insulin-inducing phenotype of artificial formulas [204].

### 6.2. Taurine

Taurine, a derivative of cysteine that contains the thiol group, is the only known natural sulfonic acid. It is classified as an amino acid but, lacking the carboxyl group, it is not strictly one. The presence of taurine has been determined in some small polypeptides. But so far, no aminoacyl tRNA synthetase, responsible for incorporating it into tRNA, has been identified [205].

It has been shown that, in newborns, the hepatic activity of certain enzymes that participate in the metabolism of taurine is limited. So an exogenous contribution of this amino acid is essential. Taurine is very abundant in breast milk. Its presence is especially relevant in excitable tissues, in cells that originate oxidizing substances, and in those organs where a large amount of toxic products are generated. It reaches particularly high concentrations in the central nervous system (even higher in newborns and during the first months of life), in the retina, and in granulocytes. Due to its osmoregulatory ability, elevated taurine concentrations in the brain, such as those seen in breastfed infants, are considered to be able to protect the nervous system from adverse effects due to both hypo and hyperosmolarity [206]. Preterm infants are more vulnerable to taurine deficiency due to low endogenous synthesis capacity and increased kidney loss [207]. It can also aid in fat

absorption through its conjugation with bile acids. In fact, children who eat enough of it, absorb fat better [208].

Breast milk has the right amount of taurine to protect brain cells from osmotic imbalances and oxidative stress. Cow's milk, the basis for formulating artificial milk, is deficient in taurine. The use of infant formulas that are poor in this nutrient has been associated with vision and hearing problems, as well as decreased bile acid secretion, reduced fat absorption, and liver cholestasis [208]. There are no studies linking the addition of taurine to formulations with problems in children, and most marketed infant formulas add taurine to breast milk levels. However, EFSA does not consider its addition necessary [6].

### 6.3. Carnitine

Carnitine, produced from the amino acids lysine and methionine, is an important and highly available component of human milk [209]. The main function of carnitine is to facilitate the transport of long-chain fatty acids through the mitochondrial membrane, facilitating their metabolism. In the newborn, it favors the use of fatty acids, inhibits neoglycogenic muscle proteolysis [210], and would have a neuroprotective effect [211]. In infants, plasma carnitine concentrations decline markedly shortly after birth [212] and this fact increases the importance of exogenous carnitine supplementation. The ESPGHAN (European Society for Pediatric Gastroenterology, Hepatology and Nutrition) has recommended since 1991 that formulas for low-weight newborns contain L-carnitine in concentrations at least similar to those in human milk [213].

### 6.4. Polyamines

They are molecules of small size, metabolically derived from certain amino acids, with a size similar to theirs, that have polycationic nature, with positive charges due to their content of ionized amino groups. Polyamines are important for the growth of various organ systems, and are also involved in the differentiation of cells of the immune system and the regulation of the response to inflammatory changes, such as immunoglobulin A levels and the relative increase in intraepithelial lymphocytes [214]. Furthermore, they are maturation factors for the small intestine as they decrease intestinal permeability to macromolecules and reduce the frequency of food allergies in children [215,216]. Human milk contains biologically active polyamines such as putrescine, spermidine, and spermine and is the first and only source of these compounds for infants [217]. These compounds are often not included in infant formulas and the concentration of polyamines is about 10 times lower than in human milk [218]. However, in animal studies, it has been observed that polyamine supplementation may resemble the effect of breastfeeding on the gastrointestinal microbiota and immune system development [219].

**Table 3.** Examining nitrogenous components of breast milk used in health applications, summary of evidences (clinical trials, cohort studies, and meta-analysis).

| Component | Utilization | Health Effects | References |
|---|---|---|---|
| GMP | Nutritional supplement | Nutritional management of phenylketonurics. Reduction of the glycemic response in adults. | [108,109] |
| Casein derived peptide, C12 | Nutraceutical | Significant decrease in systolic and diastolic blood pressure. | [111] |
| Casein derived peptide, MKP | Nutraceutical | Improves cognitive function in adults without dementia, with good tolerability and without secondary effects. | [117] |
| Casein derived peptide (Colostrinin®) | Nutraceutical | Demonstrable clinical effects, without secondary effects, in patients with Alzheimer. | [119,120,122] |
| Hydrolyzed casein (Ditriamino®) | Nutraceutical | Action against Papillomavirus. | [220] |

**Table 3.** *Cont.*

| Component | Utilization | Health Effects | References |
|---|---|---|---|
| Bovine α-lactoalbumina | Infant formula | Increases absorption of iron. Similar growth to those that were given maternal lactation. | [126,221] |
| Bovine α-lactoalbumina | Nutritional supplement | Protects la muscle mass in adults or athletes. Improvements in sleep and state of mind. | [127–130] |
| HAMLET | Pharmaceutical preparation for bladder instillation | Improved prognosis in bladder cancer. | [134] |
| HAMLET | Pharmaceutical preparation for topic application in condylomas. | Complete remission of treated injuries in more than 90% of patients treated, without significant secondary effects. | [135] |
| Bovine lactoferrin | Infant formula | Increase in absorption of iron. Decrease of bacterial sepsis risk, lesser necrotizing enterocolitis, lesser diarrhea, and lower mortality. Lower incidence of fungal sepsis in premature babies. | [152–157] |
| Human recombinant lactoferrin | Nutraceutical for children | Premature babies that receive it show a tendency to lower infectious morbidity and changes in fecal microbiome [159,160]. | [159,160] |
| Human recombinant lactoferrin | Nutraceutical for adults | Decrease in mortality of adults with severe sepsis. Increase of efficacy of H Pylori eradication therapies. Decrease of post antibiotic diarrhea in patients of advanced age. Increase of efficacy of standard interferon therapy (IFN) and ribavirin in hepatitis C and other viral infections. Reduction of symptomatology and duration of common colds. | [163–167] |
| Human recombinant lactoferrin and lysozyme | Nutraceutical for children | Treatment of acute diarrhea in children and improvement of intestinal health in general. | [222,223] |
| Hyperimmune Bovine Colostrum, HBC | Travelan® tablets | Prevents traveler's diarrhea. | [179] |
| Bovine osteopontin (Lacprodan® OPN-10) | Infant formula | Fewer episodes of fever and a less pro-inflammatory immune response. | [187,224] |
| Bile Salt Stimulated Lipase (BSSL) | Infant formula and pasteurized human milk | Improves growth rate and PUFA absorption in premature infants and small-for-gestational-age newborns. | [191,192] |
| Glutamine | Enteral supplement | Reduces invasive infection rates without affecting growth or mortality. | [225] |

## 7. Growth Factors (GFs)

Breast milk is rich in GFs, that seem to play an important role in the first moments of life, favoring the growth and development of the child. Some are found in higher concentrations in colostrum and others increase their concentration in mature milk [7].

GFs withstand the conditions of the infant's digestive system and reach the bloodstream intact, being able to reach the target organs in a bioactive form. Among them, we can mention epidermal growth factor (EGF), also present in amniotic fluid, brain-derived neurotrophic factor (BDNF), insulin-like growth factor type I (IGF-I) or transforming growth factor (TGF-β).

EGF is found in the infant's intestine, where it plays a role in intestinal maturation and repair [226]. It can limit the spread of enteric pathogens and potentially prevent systemic

infections in newborns. Furthermore, it has effects on various organs and systems through the activation of the growth factor receptor on neonatal epithelial cells [227].

BDNF is a protein that, along with another related protein, ciliary neurotropic factor (CNTF), are detected in breast milk for up to 90 days after birth. The content of neurotrophic factors and cytokines in human breast milk could influence the postnatal development of the enteric nervous system. The potent modulatory role in enteric neuronal activity and synaptic communication of BDNF is capable of enhancing enteric nervous system signaling and thus promoting intestinal peristalsis, which is often poor in the preterm infant [228].

IGF-1 factor is a single chain 70 amino acid polypeptide, is a member of a superfamily of insulin-like hormones, and acts as the main mediator of growth hormone, playing a very important role in regulating human growth [229]. IGF1 is found in human milk, in a bioactive form in the intestine of breastfed infants and in their blood serum at higher concentrations [195]. Very recent research links its concentration in human milk with an important role in defining infant growth trajectories beyond the first year of life [230] and supports previous studies that highlighted its importance in infant growth and regulation of fat accumulation during childhood [231].

The concentration of TGF-β in breast milk shows a positive correlation with the production of immunoglobulins [232], induces antigenic tolerance during colonization of the neonatal intestine [233], and attenuates the inflammatory response [234]. TGF-β has been associated with a lower risk of respiratory disease and neonatal allergy [235]. The utility of this GF in immunology has been demonstrated in experimental animal studies [235] and clinical trials [236,237]. However, in 2019, a systematic review concluded that differences in the methodology and results of the studies do not allow unconditional rejection or acceptance of the hypothesis that TGF-β influences the risk of developing allergy [238]. Its use has also been tested in Crohn's disease [239,240]. An enteral nutrition formulation is currently available for the primary treatment of pediatric Crohn's disease or as an adjunct or alternative treatment for Crohn's disease in adults (Modulen IBD, Nestlé Nutrition). The results of clinical trials in patients indicate that nutritional supplementation with TGF-β improves nutritional and inflammatory patterns (histological parameters and CRP levels) and produces an improvement in the fecal microbiome associated with disease remission [241,242], see Table 4.

## 8. Exosomes

Milk exosomes are secreted by the epithelial cells of the mammary glands and are also released from milk fat globules during lactation [243]. They are vesicles released by cells that contain various types of lipids, proteins, as well as genetic material such as microRNA [244]. The encapsulation of these molecules confers them protection against digestive degradation, and they can also be captured by cellular endocytosis and mediate the delivery of these charges to the recipient cells. The bioactive cargo molecules transporting by exosomes affect immunity, growth and development, cell proliferation, and apoptosis or differentiation of progenitor cells in the lung epithelium [245–249]. The biological and nutritional importance of the lipids and proteins of the exosomes of breast milk also remains to be discovered [250].

Although exosomes were first extracted and characterized from human colostrum and breast milk [251], they can be obtained, on a significant scale, from bovine milk and show tolerance between species [252]. For this reason, its therapeutic potential has begun to be tested in autoimmune and inflammatory diseases [253]. As natural carriers of endogenous biomolecules, milk exosomes have notable advantages over other drug delivery vehicles [249]. As they are not degraded by digestion, they could be used for the oral absorption of drugs conventionally administered intravenously such as chemotherapeutic agents [254] or as small molecules that will thereby increase their bioavailability [255]. Milk exosomes are also used for the administration of siRNA, thus avoiding its degradation by nucleases in blood serum. In this way, they are taken up by cancer cells and silence

the target genes [256]. Table 4 shows some recent clinical applications of exosomes obtained from bovine milk.

## 9. Microorganisms

In the 1920s it was already known that breast milk contained bacteria (Dudgeon and Jewesbury 1924). But until not many years ago it was assumed that the bacteria that were isolated were contaminants from the baby's oral cavity, from the mother's skin, or from utensils in contact with her. In the past decade, in addition to bacterial culture, multiple methods have been used to determine the microbiota of breast milk. Thus, with a different approach to culture, there have been used techniques such as PCR (polymerase chain reaction), PCR-DDGE (PCR-denaturing gradient gel electrophoresis), MALDI-TOF-MS (matrix assisted laser desorption ionization-time of flight mass spectrometry), 16S rRNA gene amplicon sequencing or shotgun metagenomic sequencing, in a large number of investigations. These works are allowing us to know the amazing diversity of the breast milk microbiome, which includes: potentially beneficial, commensal and probiotic bacteria [257–259]. The studies reviewed by Zimmermann and Curtis indicate more than 1300 species, with the predominant genera *Staphylococcus, Streptococcus, Lactobacillus, Pseudomonas, Bifidobacterium, Corynebacterium, and Enterococcus*, but it also contains archaea, fungi, and viruses [260].

In human milk we find a microbiota of maternal origin and another that we could consider exogenous. Regarding microorganisms of maternal origin, bacteria from the maternal gastrointestinal and oropharyngeal tract could translocate and migrate to the mammary glands through an endogenous cellular pathway (the enteric and oro-mammary tract). The diversity of the microbiota of this origin is influenced by diet, mother's lifestyle, medication, permeability of her intestinal mucosa, as well as her periodontal health. The microbiota of the mammary gland is another source of microorganisms on which mastitis, previous pregnancies, or even cancer has an influence. Regarding the exogenous origin, the existence of a retrograde translocation of bacteria from the child (which would be influenced by sex or the type of delivery) has been postulated. This exogenous source should be added to the contaminating flora of the utensils in contact with breast milk or the extractor device used in the extraction. As we can see, depending on the origin, maternal or exogenous, there will be different factors that would influence the diversity of microorganisms [261–264]. In 2020, Zimmermann and Curtis systematically summarized the factors that influence the microbiota of breast milk, concluding that: if the delivery is preterm, the Lactobacillus and *Bifidobacterium* strains increase; if the delivery is vaginal, the diversity is greater and the composition is different than when a cesarean section occurs; when it comes to children there is more diversity of genders and a decrease in *Staphylococcus* and an increase in *Streptococcus* is observed; parity increases diversity; the use of antibiotics in childbirth increases diversity but decreases the number of colony-forming units and reduces *Lactobacillus* and *Bifidobacterium*; the composition of the microbiota changes with the stage of lactation; in colostrum there are more microorganisms; Mastitis decreases the diversity of flora and changes composition, which also varies with geographic location, collection method, and type of feeding (direct to the breast or using a bottle) [260].

Today, the microbiome is considered one of the main functional assets of human milk (breastfed babies ingest up to 800,000 bacteria per day) [265]. It is the second source of microorganisms for the infant, after exposure to the vaginal and fecal maternal microbiota in the birth canal, when this occurs vaginally [266]. The transfer of bacteria from mother to child has been widely described [267] and has a strong impact on the colonization of the intestine of the breastfed infant [263,268–271]. Many epidemiological studies have documented differences in the composition of the intestinal microbiota in breastfed and formula-fed infants [272]. But it also has an impact on the colonization of its oral cavity [269] and its respiratory tract [273].

Breast milk provides a very important source of microorganisms, but also of bioactive factors that modulate the establishment of a beneficial microbiome for the present and

future health of the child [274–276]. Among these biofactors, are HMOs, determined in part by the maternal genotype [70,274]. HMOs orchestrate the development of the microbiota by playing a role in preventing pathogenic bacterial adhesion, as well as providing nutrition for microorganisms. Other components of human milk, such as exosomes (which carry a diverse load, including mRNA and microRNA), in addition to cytosolic and MFGM proteins, may also play a significant role in the development of the infant microbiome [258].

WHO recommends exclusive breastfeeding for the first six months [277]. During this time, a microbial imprint will be established that may have long-term health implications [258,266,278–281]. Indeed, strong associations have been established between whether adults had been breastfed as infants and their microbiome at various sites in the body [282]. The benefits of breastfeeding make the isolation of strains from breast milk for their subsequent use a preferential focus of attention.

Probiotic function is understood to be the ability to colonize the neonatal intestine, resist stomach acid and bile salts, adherence to the intestinal mucosa, induction of anti-inflammatory responses, inhibition of pathogens by production of antimicrobial substances, and stimulation of the immune system [283]. The potential probiotic function of the strains isolated from the maternal microbiota has been widely investigated [276,284–290].

Among the microbial colonizers of early life, mainly the strains of the genera Bifidobacterium (infantis, breve, animalis) and Lactobacillus (plantarum, rhamnosus, salivarius, reuteri, gasseri, fermentum), but also Streptococcus, Enterococcus, Bacillus, Escherichium, Propionibacterium, Lactococcus and yeasts (Saccharomyces), have been considered as probiotics with a wide range of health benefits [291–294].

Dysbiosis that lead to inflammation could be avoided if we understand the mechanisms by which the microbiota settles in the child and impacts on the child's immune development. By modulating the maternal microbiome, we could perhaps improve the microbiome of the breast milk. This and other possible interventions to improve the beneficial properties of breast milk are and will be a very important focus of research in the coming years. The existence of the entero-mammary route of microbial transfer opens the possibility of modulation of the infantile intestinal microbiota through supplementing the mother with probiotics [295]. In the ProPACT study (Probiotics in the Prevention of Allergy between Children in Trondheim), maternal supplementation during pregnancy and lactation was used to increase the prevalence and relative abundance of the probiotic strains used in the study, in the feces of the mothers and of their children, in addition to reducing by 40% the cumulative incidence of atopic dermatitis among offspring at 2 years of age [295,296]. This entero-mammary route could also explain the emerging use of *lactobacilli* strains from human milk as a prevention strategy or even as an alternative to antibiotics to treat lactational mastitis [297].

*Bifidobacterium* is the most abundant genus in the intestine of the naturally fed infant and has been considered a marker of healthy development of the microbiota [298]. Some factors such as maternal overweight [299], premature birth, cesarean delivery or early use of antibiotics [300] are known to prevent adequate colonization of the baby's intestine with bifidobacteria. For this reason and due to their symbiotic relationship with the oligosaccharides in breast milk, bifidobacteria are considered ideal probiotics for babies and are already used in currently marketed infant nutrition products.

The European Commission has favorably evaluated the addition of probiotics in infant milk, provided that its benefit and safety have been evaluated by controlled, double-blind clinical studies [301]. *B. animalis* subsp. *lactis* INL-1 [302] and *L. plantarum* 73a [293] isolated from breast milk are being tested as potential probiotics in complementary strategies for the prevention of childhood obesity [294]. These strains and others that have scientific evidence that supports their use in the prevention or improvement of different health problems, are presented in Table 4.

**Table 4.** Examining other breast milk components used in health applications, summary of evidences (clinical trials, cohort studies, and meta-analysis).

| Component | Utilization | Health Effects | References |
|---|---|---|---|
| TGF-β Modulen® IBD | Nutraceutical for children and adults | Improvement of nutritional and inflammatory patterns Crohn's disease. Changes in the fecal microbiome associated with remission of the disease. | [241,242] |
| Bovine milk exosomes | Drug delivery systems | Possibility of using intravenous drugs by mouth. Increased bioavailability of small molecules. Improvements in the effectiveness of anti-cancer drugs. Decreased off-target antitumor effects. | [303–306] |
| *B. animalis* ssp. *lactis* Bb-12 | Infant formula | Prevention of diarrhea. Improved intestinal health markers. Prevention of NEC, sepsis, and all-cause mortality among preterm infants in preterm infants. | [307–310] |
| *B. infantis* Bb-02 | Nutritional supplement | Prevention of NEC, sepsis, and all-cause mortality among preterm infants in preterm infants. | [309,310] |
| *B. breve* CECT7263 | Infant formula | Decreases the incidence of diarrhea. | [311] |
| *L. rhamnosus* GG (LGG) ATCC 53103, ATC A07FA and Lcr35 | Nutritional supplement | Reduction in the duration of infant diarrhea. Prevention of NEC, sepsis, and all-cause mortality among preterm infants in preterm infants. | [310,312] |
| *L. reuteri* DSM 17,938 and ATCC 55730 | Nutritional supplement | Effective in the prevention and treatment of infant colic. Reduction in the duration of infant diarrhea. Reduction in the frequency of respiratory infections. | [313–315] |
| *L reuteri* DSM 17938 | Infant formula | It modulates the intestinal flora in children born by cesarean section. | [316] |
| *L. fermentum* CECT5716 | Nutritional supplement for nursing mothers | Improved growth and health of the infant. Prevention and treatment of mastitis. | [317,318] |
| *L. fermentum* CECT5716 | Infant formula | Decreases the incidence of diarrhea. | [311] |
| LGG, Bb-12 and *L. acidophilus* La-5 | Nutritional supplement for pregnant and nursing mothers | Decrease in the incidence of atopic dermatitis. | [295,296] |

## 10. Concluding Remarks

Breast milk is a complex matrix that contains a large number of bioactive components, with a distribution and organization that is not accidental. The excellent tolerability and absence of side effects of compounds derived from human milk is an obvious advantage. However, the difficulty of isolating and producing these bioactive substances on an adequate scale slows down the progress toward preclinical-clinical research and their nutraceutical application. Dairy components isolated from milk of other species such as bovines, transgenic cattle, or microbial cultures are used.

The study of specific structures with clear functional activity that are present in breast milk, allows the development and/or improvement of infant formulas. To bring them as close as possible to breast milk, attempts are made to reproduce the microstructure of functional MFG emulsions. Also, formulas are enriched with components of breast milk such as β-palmitate, ARA, DHA, taurine, or carnitine. In addition, dairy fractions have

been isolated from bovine milk with bioactive components and are now commercially available. Among these components are MFGM, α-lactalbumin, lactoferrin, and osteopontin. Chemoenzymatic processes have been used in these by-products of the dairy industry to approximate the oligosaccharides of the cow and goat serum permeate to HMOs, especially in terms of fucosylated oligosaccharides. Furthermore, engineered microbial systems are currently used to produce HMO in sufficient quantities. Some strains of Bifidobacterium and Lactobacillus present in breast milk are added to artificial formulas. However, we are still a long way from replicating the complexity of human milk and many questions about its clinical implications remain unanswered. The research findings in this field should serve, above all, as another compelling reason to encourage and support breastfeeding as the "gold standard" in infant nutrition.

Apart from infant formula, breast milk bioactive compounds can be used in other food products, nutritional supplements, nutraceuticals, or they can even lead to opportunities for translational medicine. Thus, 2'FL and LNnT HMOs are included in nutritional supplements to improve intestinal flora in adults with irritable bowel syndrome. Some casein-derived peptides are incorporated into nutraceuticals to help patients with pathologies as diverse as hypertension or cognitive impairment. Bovine α-lactalbumin is used in older adults or in athlete supplements to protect muscle mass and to modify neurological or behavioral outcomes such as sleep and mood. Microbial cultures and transgenic animals are used for the large-scale production of HAMLET (α-lactalbumin lethal to tumor cells), which is used in pharmaceutical formulations against cancer. Recombinant human lactoferrin and lysozyme are used in nutraceutical preparations as antimicrobial agents with clinical efficacy in the treatment of infectious diseases in humans. HBC has been used successfully for specific antimicrobial prophylaxis in healthy adults and to treat diarrhea in children. Adding recombinant human BSSL to pasteurized breast milk or infant formula improves growth rate and absorption of long-chain PUFAs in premature infants. The TGF-β present in human milk improves the nutritional and inflammatory pat-terns of Crohn's disease. But also, exosomes from bovine milk are used as drug delivery systems. Finally, we must not forget the probiotic strains isolated from breast milk that are used in the prevention and treatment of diarrhea and mastitis.

**Author Contributions:** Conceptualization, C.S., L.F., and C.F.; methodology, C.F.; writing—original draft preparation, C.S., L.F., C.F., A.L., and P.R.; writing—review and editing, C.F., P.R. and A.C.; supervision, C.F. All authors have read and agreed to the published version of the manuscript.

**Funding:** This research received no external funding.

**Institutional Review Board Statement:** Not applicable.

**Informed Consent Statement:** Not applicable.

**Data Availability Statement:** Not applicable.

**Conflicts of Interest:** The authors declare no conflict of interest.

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
