# Peer review of "Breast Milk: A Source of Functional Compounds with Potential Application in Nutrition and Therapy"

_nutrients, doi:10.3390/nu13031026_

Round 1
Reviewer 1 Report
Review of Breast Milk: a Source of Functional Compounds with Potential Application in Nutrition and Therapy
Pg 1 line 2 – the word “title” appears in the title.
Abstract:
First two sentences do not express the thought intended and include incorrect sentence grammar.
Line 23 – “focuses”, should be ‘we focus’.
Introduction:
Line 32 – “Not only is an unbeatable food”, please include that ‘milk’ is an unbeatable food, otherwise sentence structure is confusing.
Lines 45-46 – change to ‘scientists endorse’.
Methods:
Could authors include the search criteria, key words, exclusion criteria, total number of studies found within criteria, number of studies summarized in manuscript?
Main:
Section milk fat globules - Authors do not discuss production, origins of the lipids described in various formulas. Excessive description of downstream effects, which could be better summarized, try to be more concise.
Lines 266-277 – excessive detail that is not framed within the authors’ intention of formula component production. How do companies wanting to supplement HMOs in formula take these genetic differences into account? This should be addressed if at all possible.
Lines 373-380 – Authors are not covering the scope of proteome studies performed in breast milk. A suggestion would be to remove the specific number of total proteins, differentially expressed proteins, as they vary between numerous studies.
Lines 398-401 – The relevance of ANG II and ACE are not clear in the context of the paragraph/manuscript.
Lines 419-424 – Why are authors including information on b-lactoglobulin? If authors keep this information in manuscript, it might be worthwhile to include it in the context of formula production.
Line 425 – Special character missing in title.
Tables:
Could authors include the origin of bioactive component (was it obtained from bovine milk, bacterial fermentation etc)? Could authors include short authors names in “references” section of table for ease of reading to those familiar with key authors in the field?
Overall:
A modest but unique perspective on milk components as authors probe their production for infant formulas from non-human sources. However, additional discussion on the original scope should be included in place of detailed summary description of individual components. Authors’ conclusions could include additional comparisons between non-human milk components versus those in donor milk for formula.
Could the authors add caveats or disclaimers throughout the result interpretation/relay, which take into account the variabilities between studies examined, number of infants enrolled, and other factors, which may have contributed to positive results following supplementation?
Could authors include a brief “Conclusion” section, where they detail the direction of the field, limitations, obstacles, constraints, and advances as they pertain to the original scope of the manuscript?
Grammatical and sentence errors persist throughout this draft manuscript, which confuse the overall message that the authors are attempting to convey. It would be helpful to have manuscript proof-read and corrected by a special editor dedicated to Scientific Technical English. Too many examples of bad English, two will suffice to support this critique.
Example: Second sentence is off-putting:
“Not only is an unbeatable food, … “ This introductory phrase means that there is a new baby food branded NotOnly and the sentence means, as written, that this amazing NotOnly is an excellent new food.
Example of need for more precise written epression: And this sentence in 5.2.1 does not make sense and is much too long,
“But most uses for α-lactalbumin stem from its unique amino acid composition: tryptophan, a precursor to the neurotransmitter serotonin that has been linked to central nervous system functions such as appetite, sleep, memory and learning, regulation of temperature, mood, behavior and maturation of neurons and synaptic connections; cysteine, a sulfur amino acid that stimulates the production of glutathione that plays an important role in protecting cells against oxidative stress; and the branched-chain amino acids, leucine, isoleucine and valine, stimulate postprandial anabolism of muscle proteins [131].”
Cf. “unique amino acid composition: tryptophan” - Surely the protein lactalbumin is not composed of only one single amino acid, Trp?
[end of review]
Author Response
Responses to Reviewer 1:
The authors are grateful to the referee’s comments. In the revised version of the manuscript the aspects mentioned by the reviewer have been corrected.
Pg 1 line 2 – the word “title” appears in the title. Text has been corrected.
Abstract:
First two sentences do not express the thought intended and include incorrect sentence grammar. In the revised version the abstract has been improved following the reviewer comments.
Line 23 – “focuses”, should be ‘we focus’. Text has been corrected.
Introduction:
Line 32 – “Not only is an unbeatable food”, please include that ‘milk’ is an unbeatable food, otherwise sentence structure is confusing. Text has been corrected.
Lines 45-46 – change to ‘scientists endorse’. Text has been corrected.
Methods:
Could authors include the search criteria, key words, exclusion criteria, total number of studies found within criteria, number of studies summarized in manuscript? In the revised version the Methods has been improved following the reviewer comments.
Main:
Section milk fat globules - Authors do not discuss production, origins of the lipids described in various formulas. Excessive description of downstream effects, which could be better summarized, try to be more concise. The section has been summarized. We have not discussed the different alternatives for adding MFGM to the formulas because it was not the object of the work.
Lines 266-277 – excessive detail that is not framed within the authors’ intention of formula component production. How do companies wanting to supplement HMOs in formula take these genetic differences into account? This should be addressed if at all possible. Currently the only HMOs used are 2′-fucosillactose (2′-FL) and lacto-N-neotetraose (LNnT)). Non-relevant information regarding HMOs and maternal genetics has been removed as this is not related to the subject of this work.
Lines 373-380 – Authors are not covering the scope of proteome studies performed in breast milk. A suggestion would be to remove the specific number of total proteins, differentially expressed proteins, as they vary between numerous studies. All information regarding proteins that are not used as functional components has been removed. The phrase “Following, we summarize some of these compounds used as functional components” has been included:
Lines 398-401 – The relevance of ANG II and ACE are not clear in the context of the paragraph/manuscript. We summarize some of the milk compounds used as functional components for the development and/or improvement of infant formulas, any other food product, nutritional supplements, nutraceuticals or even for the development of new clinical therapies.
Lines 419-424 – Why are authors including information on b-lactoglobulin? If authors keep this information in manuscript, it might be worthwhile to include it in the context of formula production. Information on b-albumin has been removed.
Line 425 – Special character missing in title. Text has been corrected.
Tables:
Could authors include the origin of bioactive component (was it obtained from bovine milk, bacterial fermentation etc)? Could authors include short authors names in “references” section of table for ease of reading to those familiar with key authors in the field? The origin of the components has been included. It is difficult to include the name of the authors when Nutrients uses a reference format with numbers. However, if the reviewer deems it necessary, they will be included.
Overall:
A modest but unique perspective on milk components as authors probe their production for infant formulas from non-human sources. However, additional discussion on the original scope should be included in place of detailed summary description of individual components. Authors’ conclusions could include additional comparisons between non-human milk components versus those in donor milk for formula. The conclusions section has been included. We believe it will help to understand the purpose of the work.
Could the authors add caveats or disclaimers throughout the result interpretation/relay, which take into account the variabilities between studies examined, number of infants enrolled, and other factors, which may have contributed to positive results following supplementation? Only references from clinical trials, cohort studies and meta-analysis are included in the tables. But this is not a systematic review. We are trying to show the complexity of breast milk and the possibilities offered by its bioactive components.
Could authors include a brief “Conclusion” section, where they detail the direction of the field, limitations, obstacles, constraints, and advances as they pertain to the original scope of the manuscript? The conclusions section has been included. We believe it will help to understand the purpose of the work.
Grammatical and sentence errors persist throughout this draft manuscript, which confuse the overall message that the authors are attempting to convey. It would be helpful to have manuscript proof-read and corrected by a special editor dedicated to Scientific Technical English. Too many examples of bad English, two will suffice to support this critique. Text has been improved upon grammar and sentence structure.

Reviewer 2 Report
- Please read carefully the manuscript looking for grammar and readability. In the abstract, the first two sentences is not a complete sentence (1st one) and the is grammatically incorrect. There are numerous situations like this.
- I do not find the tables helpful -- in fact they are distracting. I would find a way to present the data better -- i.e. what does 'utilization' mean (second column)? In fact, the first two were not helpful at all in my view.
- I did not find it helpful to keep comparing / going back to formula - i.e. breast milk will never compare to formula given breast milk is a moving/dynamic/changing/developing biological matrix that formula will never be, thus, not sure the comparison needs to be made.
- Hormones section -- page 21 -- it is quite sparse and incomplete - either delete it or expand it i.e. no insulin (there is a fair bit of data on that) and only one paper (you have cited) for adiponectin, leptin (has 3) and ghrelin (has 3) there are many many more papers than that. And HMO's there are many keystone papers that are missing.
- It is my view that this review would be best served by focusing a bit on what you are looking at vs. EVERYTHING --- you have well over 300 citations now, and many many are missing -- thus, I think the paper is greatly improved if you narrow your focus.
Author Response
Responses to Reviewer 1:
The authors are grateful to the referee’s comments. In the revised version of the manuscript the aspects mentioned by the reviewer have been corrected.
- Please read carefully the manuscript looking for grammar and readability. In the abstract, the first two sentences is not a complete sentence (1st one) and the is grammatically incorrect. There are numerous situations like this.
Text has been improved upon grammar and sentence structure.
- I do not find the tables helpful -- in fact they are distracting. I would find a way to present the data better -- i.e. what does 'utilization' mean (second column)? In fact, the first two were not helpful at all in my view. The authors consider the tables illustrative on the variety of use of bioactive ingredients in breast milk.
- I did not find it helpful to keep comparing / going back to formula - i.e. breast milk will never compare to formula given breast milk is a moving/dynamic/changing/developing biological matrix that formula will never be, thus, not sure the comparison needs to be made.
The authors do not try to compare the artificial formula with breast milk, in fact we consider that they are not comparable. This article aims to review the biactives present in breast milk that can be incorporated into artificial milk to bring it closer to breast milk. In conclusions, the phrase has been included: However, we are still a long way from replicating the complexity of human milk and many questions about its clinical implications remain unanswered. The research findings in this field should serve, above all, as another compelling reason to encourage and support breastfeeding as the “gold standard” in infant nutrition.
- Hormones section -- page 21 -- it is quite sparse and incomplete - either delete it or expand it i.e. no insulin (there is a fair bit of data on that) and only one paper (you have cited) for adiponectin, leptin (has 3) and ghrelin (has 3) there are many many more papers than that. And HMO's there are many keystone papers that are missing. The hormones section has been removed. According to the reviewer, we think it is outside the scope of this review. In this article we have reviewed the bioactive compounds that are used as functional ingredients.
- It is my view that this review would be best served by focusing a bit on what you are looking at vs. EVERYTHING --- you have well over 300 citations now, and many many are missing -- thus, I think the paper is greatly improved if you narrow your focus. According to the referee, we have eliminated all the compounds that are not currently being used in infant formulas, in other food products, nutritional supplements, nutraceuticals, or they can even lead to opportunities for translational medicine.

Round 2
Reviewer 1 Report
Reviewer has looked through the updated manuscript and it seems that the co-authors have definitely made some improvements as suggested. Though, there are still typos and illogical sentence structure throughout. You will benefit greatly from engaging a professional English science writer. As per my original comments on reading the very first submitted draft, they consortium of co-authors should have written a whole new paper to improve the sentence structure/logic.
But...As an overall literature review, this manuscript seems to achieve its purpose. The Editor-in-Chief should decide on the expected professional standard of written English in this case.